# Konjac Glucomannan Attenuated Triglyceride Metabolism during Rice Gruel Tolerance Test

**DOI:** 10.3390/nu13072191

**Published:** 2021-06-25

**Authors:** Takumi Nagasawa, Takao Kimura, Akihiro Yoshida, Katsuhiko Tsunekawa, Osamu Araki, Kazumi Ushiki, Hirotaka Ishigaki, Yoshifumi Shoho, Itsumi Suda, Suguru Hiramoto, Masami Murakami

**Affiliations:** 1Department of Clinical Laboratory Medicine, Gunma University Graduate School of Medicine, Showa-machi 3-39-22, Meabshi 371-8511, Gunma, Japan; t_nagasawa@gunma-u.ac.jp (T.N.); ayossie10@gunma-u.ac.jp (A.Y.); ktsune@gunma-u.ac.jp (K.T.); oaraki@gunma-u.ac.jp (O.A.); kazumi.15@gunma-u.ac.jp (K.U.); ishigaki@paz.ac.jp (H.I.); shoho@ikuei-g.ac.jp (Y.S.); isuda@gunma-u.ac.jp (I.S.); s-hiramoto@gunma-u.ac.jp (S.H.); mmurakam@gunma-u.ac.jp (M.M.); 2Clinical Laboratory Center, Gunma University Hospital, Showa-machi 3-39-22, Meabshi 371-8511, Gunma, Japan; 3Center for Food Science and Wellness, Gunma University, Aramaki-machi 4-2, Meabshi 371-8510, Gunma, Japan

**Keywords:** dietary supplements, konjac glucomannan, rice gruel, triglyceride, lipoprotein lipase (LPL), glycosylphosphatidylinositol-anchored high-density lipoprotein-binding protein 1 (GPIHBP1), hepatic triglyceride lipase (HTGL)

## Abstract

In a recent study, we showed that konjac glucomannan (KGM) inhibits rice gruel-induced postprandial increases in plasma glucose and insulin levels. To extend this research, we investigated the effects of KGM addition to rice gruel on pre- and postprandial concentrations of circulating lipoprotein lipase (LPL), glycosylphosphatidylinositol-anchored high-density lipoprotein-binding protein 1 (GPIHBP1), hepatic triglyceride lipase (HTGL), free fatty acids (FFA), and triglycerides (TG). A total of 13 Japanese men, without diabetes, dyslipidemia, or gastrointestinal diseases, interchangeably ingested rice gruel containing no KGM (0%G), rice gruel supplemented with 0.4% KGM (0.4%G), and rice gruel supplemented with 0.8% KGM (0.8%G), every Sunday for 3 weeks. Blood samples were obtained at baseline and at 30, 60, and 120 min after ingestion to measure the abovementioned lipid parameters. Lipid parameters showed small, but significant, changes. Significant reductions were found in circulating FFA levels among all participants. Circulating TG levels significantly declined at 30 min and then remained nearly constant in the 0.8%G group but exhibited no significant difference in the 0%G and 0.4%G groups. Although circulating levels of LPL and GPIHBP1 significantly decreased in the 0%G and 0.4%G groups, they increased at 120 min in the 0.8%G group. Participants in the 0%G and 0.4%G groups showed significant decreases in circulating HTGL levels, which was not observed in the 0.8%G group. Our results demonstrate the novel pleiotropic effects of KGM. Supplementation of rice gruel with KGM powder led to TG reduction accompanied by LPL and GPIHBP1 elevation and HTGL stabilization, thereby attenuating TG metabolism.

## 1. Introduction

Konjac glucomannan (KGM) is a rich source of soluble fiber that contains almost no calories [1]. Because KGM has good biocompatibility and biodegradability, it is widely used globally as a medicinal supplement, and is also used as a traditional food in Japan in the form of konjac jelly, noodles, and tofu [1,2,3,4,5,6,7]. KGM has a significant role in the treatment and prevention of various diseases because of its pleiotropic effects, including anti-diabetic, anti-obesity, anti-inflammatory, prebiotic, antibiotic, immune regulatory, and laxative activities. Previous studies have shown that KGM offers major health benefits—such as lowering blood glucose, cholesterol, triglyceride (TG), lowering blood pressure levels, and reducing body weight by promoting intestinal activity and boosting immune function in humans [1,2,3,4,5,6,7]. In these investigations, participants habitually ingested KGM every day for 8 weeks [1,2,3,4,5,6,7]. KGM absorbs digested body waste in the stomach and intestines by entrapping it within a gelatinous mass, which is eliminated from the body without being absorbed. In fact, due to its characteristic of becoming viscous in the gastrointestinal tract, KGM acts as a barrier to the absorption of sugars and nutrients [1,2,3,4,5,6,7].

In Japan, KGM is abundantly available, easily accessible, and commonly incorporated into various food products. However, despite its considerable popularity in Japan, only a few studies have demonstrated its impact on the immediate suppression of postprandial glucose, insulin, and lipid levels, in individuals with normal or impaired glucose tolerance. Our previous study was the first to show that the intake of KGM-supplemented rice gruel could suppress postprandial increase in circulating glucose and insulin levels [8]. To expand on our previous research, we investigated immediate effects of KGM on postprandial TG, their associated enzyme called lipoprotein lipase (LPL), and a related glycolipid-anchored protein known as glycosylphosphatidylinositol-anchored high-density lipoprotein-binding protein 1 (GPIHBP1). LPL is the central molecule in plasma lipid metabolism, hydrolyzing TG within TG-rich lipoproteins (TRLs), and releasing lipid nutrients for vital tissues [9]. Genetic variation, altering the efficiency of LPL-mediated TRL processing, influences both plasma TG levels and the risk for coronary heart disease [10,11]. GPIHBP1 is solely responsible for capturing LPL within interstitial spaces and shuttling it across endothelial cells to its site of action in the capillary lumen [12]. GPIHBP1-bound LPL is required for the margination of TRLs along capillaries, allowing the lipolytic processing of TRLs to proceed [13]. Missense mutations that interfere with LPL–GPIHBP1 interactions profoundly impair intravascular TG processing, resulting in severe hypertriglyceridemia [14,15]. As it is a simple and easy-to-use marker that can be applied in clinical practice, fasting LPL mass may prove useful in predicting postprandial TG elevation in fasting individuals with normal TG levels [16]. Alternatively, no significant increase in LPL activity was found during chylomicron, and very low-density lipoprotein (VLDL) overload after eating different kinds of food [17]. These findings lead us to explore food ingredients, such as KGM, that may increase circulating LPL levels. Hence, this study aimed to examine pre- and post-load TG, LPL, and GPIHBP1 levels among individuals receiving rice gruel with or without KGM.

## 2. Materials and Methods

### 2.1. Participants

This study performed additional analysis of our earlier study, which demonstrated that the intake of KGM-supplemented rice gruel inhibited postprandial increases in circulating glucose and insulin levels [8]. Written informed consent was obtained from all participants, and the Gunma University Ethical Review Board for Medical Research Involving Human Subjects approved the study protocol (UMIN registration number: UMIN000025950). This study follows the rules of the Declaration of Helsinki. As shown in Figure 1, a total of 24 middle-aged Japanese men without diabetes mellitus (as confirmed by a 75 g oral glucose tolerance test based on the guidelines of The Japan Diabetes Society [18]) underwent a rice gruel tolerance test. Eleven of these men were excluded because of dyslipidemia, and the remaining 13 were included in the study. Baseline characteristics of the included subjects are presented in Table 1. None of the participants had diabetes, gastrointestinal diseases, or a history of statin treatment (Figure 1). 

### 2.2. Gruel Test

We prepared three types of gruel (GREEN LEAF Co., Ltd., Showa, Gunma, Japan): KGM-free rice gruel (0%G), rice gruel containing 0.4% KGM powder (0.4%G), and rice gruel supplemented with 0.8% KGM powder (0.8%G). All three types of rice gruel weighed 250 g, with 0%G, 0.4%G, and 0.8%G containing 75, 77, and 80 kcal, respectively [8]. Because of the difficulty in dissolving KGM powder into rice gruel, the manufacturer (GREEN LEAF Co., Ltd., Showa, Gunma, Japan) developed the original method to make rice gruel containing KGM powder. Each type of rice gruel was packed at exactly 250 g by the manufacturer.

### 2.3. Study Design

A double-blind, randomized, crossover design was used, in which all participants underwent a rice gruel tolerance test involving ingestion of 0%G, 0.4%G, or 0.8%G every Sunday for 3 weeks (Figure 1). To perform a double-blind randomized trial, we randomly assigned participants to 6 groups with different experimental protocols (Figure 1). All participants performed a rice gruel tolerance test after a 12-h overnight fast. All participants started eating rice gruel at 8:40 and ate within 10 min. Blood sampling was performed at baseline and at 30, 60, and 120 min after rice gruel ingestion. We measured TG, free fatty acid (FFA), LPL, GPIHBP1, and hepatic TG lipase (HTGL) concentrations in the collected blood samples. Circulating levels of fasting high-density lipoprotein cholesterol (HDL-C), low-density lipoprotein cholesterol (LDL-C), TG, and hemoglobin A1c (HbA1c) were also measured at baseline.

### 2.4. Laboratory Assays

Circulating levels of HDL-C, LDL-C, TG, and FFA were measured using enzymatic methods (LABOSPECT 008, Hitachi, Tokyo, Japan). Serum LPL concentrations were determined using an LPL assay kit (SEKISUI MEDICAL Co. Ltd., Tokyo, Japan) based on a sandwich enzyme-linked immunosorbent assay (ELISA) [19]. A human GPIHBP1 assay kit and a human serum HTGL ELISA kit (Immuno-Biological Laboratories Co., Ltd., Gunma, Japan) were used to estimate serum GPIHBP1 and HTGL concentrations, respectively [19]. Plasma glucose concentrations were determined using a hexokinase method (ADAMS Glucose GA-1170; Arkray, Kyoto, Japan), and HbA1c levels were measured using high-performance liquid chromatography (ADAMS A1c HA8180; Arkray, Kyoto, Japan). Serum insulin concentrations were measured using chemiluminescence immunoassay (AIA-2000 LA; Tosoh, Tokyo, Japan) [8,19].

### 2.5. Statistical Analysis

Statistical analyses were performed using IBM SPSS Statistics, Version 25.0 (Armonk, NY, USA). Because almost all variables were not normally distributed, data are expressed as median values with a 25th–75th percentile range, rather than as mean values with standard deviations. The effects of time on TG, LPL, GPIHBP1, and HTGL concentrations (i.e., changes from baseline) were analyzed by the Wilcoxon signed-rank test to identify statistically significant differences, as appropriate. Statistical significance was set at a *p*-value of <0.05. Spearman’s correlation analysis was conducted to identify the association between circulating levels of LPL, GPIHBP1, or HTGL and analytics. Differences and correlations were considered significant when the *p*-value was <0.05. 

Trial registration: UMIN registration number: UMIN000025950; registered on 1 February 2017; https://upload.umin.ac.jp/cgi-open-bin/ctr_e/ctr_view.cgi?recptno=R000029247, accessed on 24 June 2021. 

## 3. Results

### 3.1. Changes in Circulating TG Levels

Baseline characteristics of the included subjects are presented in Table 1.

There was a significant reduction in circulating FFA levels after ingesting all three types of rice gruel. The serum levels of FFA in all participants were significantly lower at 30, 60, and 120 min after ingestion than at baseline (Figure 2A). No significant differences in circulating FFA levels were observed between the 0.8%G, 0.4%G, and 0%G groups at 0, 30, 60, and 120 min after ingestion (Figure 2A). Changes in circulating levels of TG in participants who received 0.8%G were small, but significant. Circulating TG levels in participants who received 0.8%G were significantly lower at 30, 60, and 120 min after ingestion than at 0 min (Figure 2B). Subjects in the 0%G, 0.4%G, and 0.8%G groups did not significantly differ in terms of circulating TG levels at baseline (Figure 2B). These results indicate that supplementation of rice gruel with KGM was significantly associated with lower circulating TG levels compared with fasting TG levels (Figure 2B). Although we already showed that fasting blood glucose and insulin showed narrow distribution [8], circulating levels of TG at baseline exhibited a wide distribution (Figure 2B). These findings might have been caused by a difficulty in managing eating habits during the study and a different mechanism regulating glucose and lipid metabolism. Experiments were conducted at participants’ homes, and they ate each domestic meal at home during the 3-week study period.

### 3.2. Changes in Circulating LPL, GPIHBP1, and HTGL Levels

Circulating levels of LPL, GPIHBP1, and HTGL showed small but significant changes. Circulating LPL levels in the 0%G group were significantly lower at 30 and 120 min after ingestion than at 0 min (Figure 2C). Participants in the 0.4%G group had significantly lower circulating LPL levels at 30 and 60 min than at baseline (Figure 2C). In contrast, circulating LPL levels in the 0.8%G group were significantly higher at 120 min than at 0 min (Figure 2C). The circulating levels of LPL at baseline were significantly higher in the 0%G and 0.4%G groups than in the 0.8%G group (Figure 2C). In all groups, significantly lower circulating levels of GPIHBP1 were observed at 30 min than at 0 min (Figure 2D). In contrast, circulating GPIHBP1 levels in the 0.8%G group were significantly higher at 120 min after ingestion than at 0 min (Figure 2D). Subjects in the 0%G and 0.4%G groups displayed significantly higher baseline circulating levels of GPIHBP1 than those in the 0.8%G group (Figure 2D). In the 0%G group, circulating levels of HTGL were significantly lower at 30, 60, and 120 min after ingestion than at 0 min (Figure 2E). Participants in the 0.4%G group had significantly lower circulating levels of HTGL at 30 min than at 0 min (Figure 2E). Baseline circulating levels of HTGL in the 0%G and 0.4%G groups were significantly higher than those in the 0.8%G group (Figure 2E). Baseline circulating levels of HTGL were significantly lower in the 0.4%G than in the 0.8%G group (Figure 2E). In the 0.8%G group, no significant difference was detected between circulating HTGL levels at 30, 60, and 120 min and at 0 min (Figure 2E).

### 3.3. Circulating Levels of Fasting HTGL and FFA Were Significantly Correlated

To analyze associations among circulating levels of fasting TG, FFA, LPL, GPIHBP1, and HTGL, we combined the baseline data pertaining to the 0%G, 0.4%G, and 0.8%G groups. Therefore, the number of participants was 39 (Table 2). We found a significant negative correlation between circulating levels of fasting FFA and HTGL (Table 2). 

## 4. Discussion

This study investigated the effects of KGM supplementation on postprandial changes in circulating levels of lipid parameters, including TG, FFA, LPL, GPIHBP1, and HTGL, among middle-aged Japanese participants. Our results show that the intake of rice gruel supplemented with KGM powder led to a decrease in circulating TG levels, elevated LPL/GPIHBP1 levels, and stable HTGL concentrations. To the best of our knowledge, these findings have not been previously reported. 

Our previous study showed that the intake of rice gruel containing KGM powder suppressed postprandial increases in both plasma glucose and insulin concentrations [8], which could be explained by reduced glucose absorption as a result of KGM supplementation. However, the findings of the current study cannot be attributed to inhibition of lipid absorption.

Previous human and animal studies reported that habitual supplementation with KGM lowers plasma glucose, TG, and cholesterol levels [1,2,3,4,5,6,7,20,21,22,23,24,25,26]. The hypolipidemic effect of KGM supplementation has been explained by two mechanisms [1,2,3,4,5,6,7,20,21,22,23,24,25,26]. In the first mechanism, KGM absorbs digested body waste in the stomach and intestines by entrapping it within a gelatinous mass, which is subsequently eliminated from the body without being absorbed. The gelatinous nature of KGM in the body provides a sense of satiety and fullness, and promotes peristalsis, thus regularizing bowel movements. In fact, KGM acts as a barrier to the absorption of sugars and nutrients by becoming viscous in the gastrointestinal tract [1,2,3,4,5,6,7,20,21,22,23,24,25,26]. It has also been reported that KGM supplementation inhibits cholesterol absorption in the jejunum [22] and bile acid absorption in the ileum [23], thereby improving serum lipid regulation [21]. The second mechanism underlying the hypolipidemic effect of KGM supplementation involves inhibition of hepatic cholesterol synthesis. Soluble fibers were fermented by bacteria in the colon, forming gases and short-chain fatty acids. These short-chain fatty acids were almost completely absorbed into the portal vein and may affect hepatic cholesterol synthesis [27]. The present study proposes a third mechanism to explain the hypolipidemic effect of KGM, which involves TG reduction through increased levels of LPL and GPIHBP1 and sustained levels of HTGL. Intravascular processing of TRLs by the LPL–GPIHBP1 complex is crucial for clearing TG from the bloodstream and for the delivery of lipid nutrients to vital tissues [9,10,11,12,13,14,15,28]. Supplementation of rice gruel with 0.8% KGM slightly, but significantly, reduces circulating TG levels at 30, 60, and 120 min after ingestion. This phenomenon was accompanied by a small increase in the circulating levels of LPL and GPIHBP1 and sustained levels of HTGL. Previously, heparin pretreatment was needed to analyze circulating levels of LPL and HTGL. Recently, we developed a highly sensitive ELISA to determine the circulating levels of LPL, GPIHBP1, and HTGL in blood samples that had not been pretreated with heparint [29,30,31,32]. Thereafter, we were able to detect small and significant changes in LPL, GPIHBP, and HTGL [29,30,31,32]. In the conditions of this study, heparin pretreatment was not used, and although circulating levels of LPL, GPIHBP1 and HTGL showed only small changes, these changes were statistically significant. These results indicate that the reduction in circulating TG levels in the 0.8%G group can be partially explained by LPL and GPIHBP1 elevation. However, GPIHBP1 is a membrane anchored protein, and there is no evidence that the level of circulating GPIHBP1 acts as a marker for levels of functional GPIHBP1. In this study, we could not reveal the mechanism which regulates the circulating levels of LPL, GPIHBP1, and HTGL. 

Additionally, this study suggests that fasting circulating levels of TG, LPL, GPIHBP1, and HTGL might be less stable than those of glucose and insulin. In our previous study, fasting levels of glucose and insulin in the 0%G, 0.4%G and 0.8%G groups showed narrow distribution and no significant differences [8], while circulating TG levels at baseline showed wide distribution in the 0.4%G and 0.8%G group. Similarly, circulating LPL levels at baseline in the 0%G group showed wide distribution. Circulating LPL levels at baseline in the 0.8%G group were significantly lower than those of the 0%G and 0.4%G group. These results suggest that among subjects within the physiologically normal range of lipid metabolism, eating behavior affects fasting levels of TG and LPL, but not glucose and insulin levels. A similar effect was shown in circulating levels of GPIHBP1 and HTGL at baseline. This suggests the existence of an unknown mechanism regulating TG levels, independent of glucose metabolism.

A reciprocal relationship between intracellular lipolysis and the efficiency of LPL-mediated fat storage has been demonstrated in previous studies [33,34]. In the fed state, intracellular lipase activity is low, and there is a fatty acid gradient from outside toward inside, promoting efficient uptake of LPL-generated fatty acids into the adipose tissue, where they are subsequently directed toward esterification and ultimately stored as TG. In the fasting state, however, intracellular lipase activity is higher and the gradient tends to be outward, resulting in TG fatty acid spillover [33,34]. Consistent with previous research, the fed state induced by the intake of rice gruel, with or without KGM, causes a significant reduction in circulating FFA levels. In the 0.8%G group, the reduction in circulating TG levels is accompanied by elevated LPL and GPIHBP1 levels, as well as sustained steady HTGL concentrations, which can be partially explained by the efficient uptake of LPL-generated fatty acids during the fed state [33,34]. This study suggests that there is a possibility that KGM supplementation sustains efficient uptake of LPL-generated fatty acids over 120 min. Conversely, no significant increase in LPL activity has been reported during chylomicron and VLDL overload after ingestion of different kinds of food [17]. These discrepancies might have arisen from differences in blood sampling time or test meals. KGM is a rich source of soluble fiber [1,2,3,4,5,6,7] and regulates gastrointestinal and bowel movements owing to its viscosity and gelatinous nature [1,2,3,4,5,6,7].

In the present study, circulating HTGL levels declined significantly in the 0%G group but remained unchanged in the 0.8%G group during the test period. This is in consensus with the findings of a study by Kagawa et al. [35], which showed the rapid clearance of serum TG by activated HTGL. Hence, it can be suggested that the observed reduction in circulating TG levels in the 0.8%G group is partially due to the sustained circulating levels of HTGL.

A previous study has reported no change in HTGL activity after an infusion of lipid emulsions [36]. Plasma concentration of HTGL remains in a fairly static equilibrium with its corresponding vascular binding sites, and shifts in metabolic conditions do not rapidly change HTGL activity or its interactions with these vascular sites [17]. Additionally, Kagawa et al. have demonstrated that hepatic FFA levels change in line with HTGL activation [36]. In the present study, a significant negative correlation was identified between fasting levels of FFA and HTGL. These results imply that supplementation of rice gruel with KGM could reduce circulating TG levels through LPL and GPIHBP1 elevation and HTGL stabilization. However, no underlying mechanisms which mediate LPL and GPIHBP1 elevation and HTGL stabilization after KGM ingestion were revealed.

In Japan, KGM is commonly incorporated into various food products. However, according to our unpublished data, there are seasonal fluctuations in the intake of KGM by the Japanese. KGM intake in Japan is high in winter and low in summer. This study was conducted in the summer therefore it is not necessary to consider daily intake of KGM in this study.

This study has some limitations. First, only a small number of subjects were examined. Second, this study did not control eating habits during the study period. Third, this study did not assess the levels of other TG-related markers (e.g., chylomicron) or the activity of LPL and HTGL. Fourth, this study investigated only transient beneficial effects of KGM supplementation into rice gruel. Therefore, it is still uncertain that KGM supplementation for more than 2 weeks induces elevation of LPL, GPIHBP1, and HTGL. Fifth, the mechanism which regulates circulating levels of LPL, GPIHBP1, and HTGL was not revealed. Thus, the current study is preliminary, and its limitations should be addressed in future research.

In conclusion, our results reveal novel pleiotropic effects of KGM. Supplementation of rice gruel with KGM powder can lead to TG reduction accompanied by LPL and GPIHBP1 elevation and HTGL stabilization, thereby attenuating lipid metabolism. Further study is needed to elucidate the regulatory mechanism underlying these effects.

## Figures and Tables

**Figure 1 nutrients-13-02191-f001:**
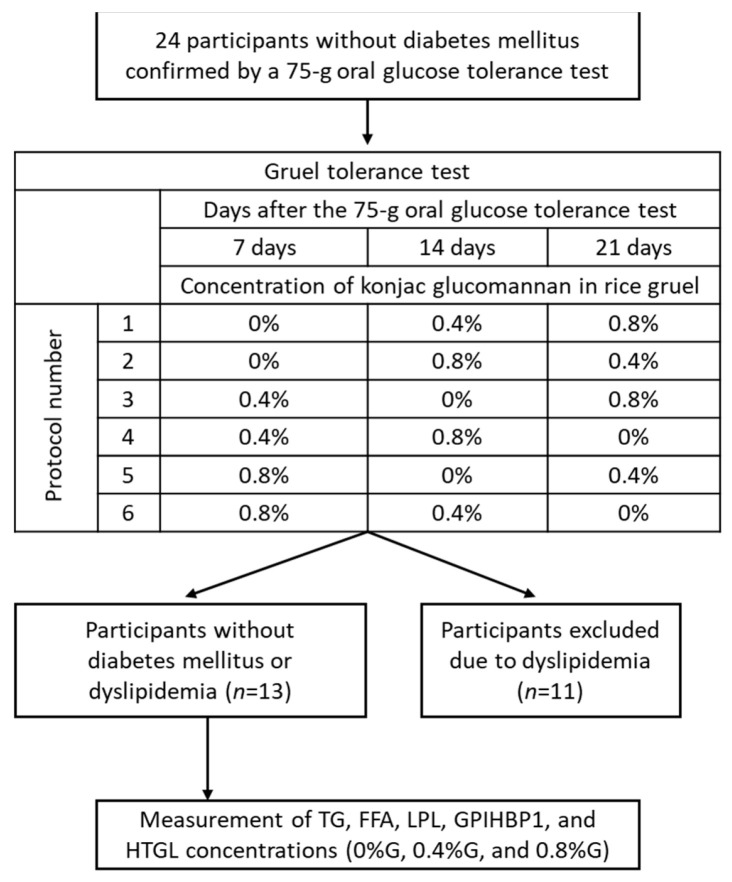
Study design. In total, 24 subjects without a history of diabetes mellitus underwent a gruel tolerance test, which involved ingesting rice gruel supplemented with konjac glucomannan at 3 different concentrations (0%, 0.4%, and 0.8%) within 3 weeks. Eleven participants were diagnosed with dyslipidemia and were consequently excluded. Finally, 13 subjects were enrolled in the study. FFA, free fatty acid; GPIHBP1, glycosylphosphatidylinositol-anchored high-density lipoprotein-binding protein 1; HTGL, hepatic triglyceride lipase; LPL, lipoprotein lipase; TG, triglyceride.

**Figure 2 nutrients-13-02191-f002:**
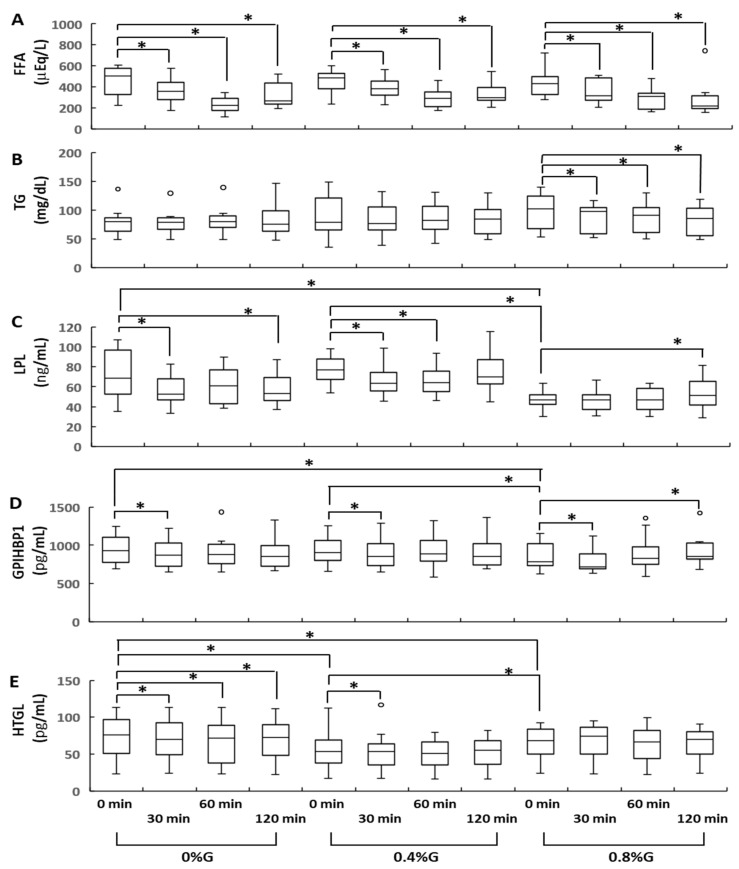
Sequential changes in the serum concentrations of FFA (**A**), TG (**B**), LPL (**C**), GPIHBP1 (**D**), and HTGL (**E**) during the gruel tolerance test. 0%G, rice gruel containing no Konjac glucomannan powder; 0.4%G, rice gruel supplemented with 0.4% Konjac glucomannan powder; 0.8%G, rice gruel supplemented with 0.8% Konjac glucomannan powder. Data are presented as median values with a 25th–75th percentile range. * indicates statistical significance (*p* < 0.05). FFA, free fatty acid; GPIHBP1, glycosylphosphatidylinositol-anchored high-density lipoprotein-binding protein 1; HTGL, hepatic triglyceride lipase; LPL, lipoprotein lipase; TG, triglyceride.

**Table 1 nutrients-13-02191-t001:** Baseline characteristics of the 13 subjects without diabetes mellitus or dyslipidemia.

	All Participants (*n* = 13)
Age (years)	47 (41–50)
Weight (kg)	73.1 (70.3–80.0)
BMI (kg/m^2^)	24.9 (23.2–26.5)
Systolic blood pressure (mmHg)	125.0 (120–134.8)
Diastolic blood pressure (mmHg)	77.0 (73.5–80.5)
Fasting plasma glucose (mg/dL)	105.0 (100–107.8)
Fasting plasma insulin (U/mL)	5.0 (4.3–6.0)
HbA1c (%)	5.5 (5.3–5.6)
LDL-C (mg/dL)	114 (89.3–134.8)
HDL-C (mg/dL)	50.0 (41.0–56.0)
TG (mg/dL)	73.0 (61.3–87.5)
FFA (μEq/L)	506.0 (330–559)
LPL (ng/mL)	69 (52.9–94.0)
GPIHBP1 (pg/mL)	930.0 (778.3–1020.6)
HTGL (pg/mL)	76.0 (51.5–96.6)

Data are expressed as medians (25th–75th percentile range). BMI, body mass index; HbA1c, hemoglobin A1c; LDL-C, low-density lipoprotein cholesterol; HDL-C, high-density lipoprotein cholesterol; TG, triglyceride; FFA, free fatty acids; LPL, lipoprotein lipase; GPIHBP1, glycosylphosphatidylinositol-anchored high-density lipoprotein-binding protein 1; HTGL, hepatic triglyceride lipase.

**Table 2 nutrients-13-02191-t002:** Spearman’s correlation analysis of baseline FFA, TG, LPL, GPIHBP1, and HTGL concentrations at baseline.

	FFA		TG		LPL		GPIHBP1		HTGL	
	*r*	*p*	*r*	*p*	*r*	*p*	*r*	*p*	*r*	*p*
FFA			−0.094	0.569	0.193	0.240	0.149	0.364	−0.358	0.025 *
TG	−0.094	0.569			−0.238	0.144	−0.269	0.098	0.069	0.677
LPL	0.193	0.240	−0.238	0.144			0.142	0.389	−0.115	0.487
GPIHBP1	0.149	0.364	−0.269	0.098	0.142	0.389			0.015	0.930
HTGL	−0.358	0.025 *	0.069	0.677	−0.115	0.487	0.014	0.930		

FFA, free fatty acids; TG, triglycerides; LPL, lipoprotein lipase; GPIHBP1, glycosylphosphatidylinositol-anchored high-density lipoprotein-binding protein 1; HTGL, hepatic triglyceride lipase. * indicates statistical significance (*p* < 0.05).

## Data Availability

The datasets generated during and/or analyzed during the current study are available from the corresponding author on reasonable request. All data generated or analyzed during this study are included in this published article.

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
