# Peer review of "Konjac Glucomannan Attenuated Triglyceride Metabolism during Rice Gruel Tolerance Test"

_nutrients, 2021, doi:10.3390/nu13072191_

Round 1

Reviewer 1 Report

The authors have carried out a suitable study design to test postprandial KGM effectors on lipid metabolism. In this sense, the study is well designed, but the authors need to improve some points  of the  material and methods and discussion section.

Materials and methods:

Please, add information about the hour of the day that subjects ate the rice and the amount of it. Was it the same for everyone? What criteria have been followed to administer the portion of rice tested?

Discussion:

First, they should expand the discussion on the limitations of the study. Indeed it is a punctual intake with postprandial analysis and without controlling the diet, at least the day before or the same day. They should explain why they have not controlled the diet during the days of KGM intake. And also what they believe that it would happen if the ingestion of KCM was carried out on a frequent daily basis. Rice is a common food of the Japanese diet, so the effects of rice with KCM should have been studied within the normal daily diet and maintained over time. Authors should expand on this part of the discussion.

Author Response

The authors have carried out a suitable study design to test postprandial KGM effectors on lipid metabolism. In this sense, the study is well designed, but the authors need to improve some points of the material and methods and discussion section.

Materials and methods:

Please, add information about the hour of the day that subjects ate the rice and the amount of it. Was it the same for everyone? What criteria have been followed to administer the portion of rice tested?

Answer: We deeply appreciate the reviewers' comments. All participant ate rice gruel at the same time. Because of the difficulty of dissolving KGM powder into rice gruel, the manufacturer (GREEN LEAF Co., Ltd., Showa, Gunma, Japan) developed the original method to make rice gruel containing KGM powder. Each rice gruel, 0%G, 0.4%G and 0.8%G was packed exactly 250g by the manufacturer. Therefore, each rice gruel, 0%G, 0.4%G and 0.8%G was same for all participant. To explain these facts, we added several sentences as described below.

  1. We added the sentence described as follows in 3 Study design (line 108-110 page 3).

“All participants performed a rice gruel tolerance test after a 12-h overnight fast. All participants started eating rice gruel at 8:40 and ate within 10 minutes.”

  1. We added the sentence described as follows in2 Gruel test (line 99-103 page 3).

“Because of the difficulty of dissolving KGM powder into rice gruel, the manufacturer (GREEN LEAF Co., Ltd., Showa, Gunma, Japan) developed the original method to make rice gruel containing KGM powder. Each rice gruel containing 0%, 0.4% and 0.8% of KGM powder was packed exactly 250g by the manufacturer.”

Discussion:

First, they should expand the discussion on the limitations of the study. Indeed it is a punctual intake with postprandial analysis and without controlling the diet, at least the day before or the same day. They should explain why they have not controlled the diet during the days of KGM intake. And also what they believe that it would happen if the ingestion of KCM was carried out on a frequent daily basis. Rice is a common food of the Japanese diet, so the effects of rice with KCM should have been studied within the normal daily diet and maintained over time. Authors should expand on this part of the discussion

Answer: We deeply appreciate the reviewers' comments. We would like to explain our opinion. Please confirm our answers.

  1. Previous reports showed positive effects of KGM by long term habitual ingestion of KGM. We intended to investigate immediate effects of KGM on postprandial TG metabolism in this study. Therefore, we performed rice gruel tolerance test after a 12-h overnight fast. To explain the purpose of this study, we described several sentences in Introduction.

Introduction (line 47-51 page 2), “Previous studies have shown that KGM offers major health benefits—such as lower blood glucose, cholesterol, triglyceride (TG), and blood pressure levels as well as reduced body weight—by promoting intestinal activity and boosting immune function in humans [1-7]. In these investigations, participants habitually ingested KGM every day for 8 weeks [1-7]”

Introduction (ine 57-64 page 2), “only a few studies have demonstrated its impact on immediate suppression of postprandial glucose, insulin, and lipid levels in individuals with normal or impaired glucose tolerance. Our previous study was the first to show that the intake of KGM-supplemented rice gruel could suppress postprandial increase in circulating glucose and insulin levels [8]. Expanding our previous research, we investigated immediate effects of KGM on postprandial TG, their associated enzyme called LPL, and a related glycolipid-anchored protein known as glycosylphosphatidylinositol-anchored high-density lipoprotein-binding protein 1 (GPIHBP1).”

  1. We added the sentence in 3 Study design, line 108-109 page 3.

“All participants performed a rice gruel tolerance test after a 12-h overnight fast.”

  1. We performed the rice gruel tolerance test after a 12-h overnight fast. We did not control daily eating behavior. Although we did not control eating behavior, fasting levels of glucose and insulin showed stable and narrow range (our previous study, ref. 8), while fasting levels of TG, LPL, GPIHBP1 and HTGL showed wide range. Both our previous study and this study analyzed the same blood sample. These results suggested that among subjects within physiologically normal range of lipid metabolism, eating behavior affected fasting levels of TG and LPL, but not glucose and insulin. Similar effect was shown in circulating levels of GPIHBP1 and HTGL at baseline. To expand these discussion, we added the sentence described as follows in Discussion (line 287-290 page 8).

The former sentence: “These results suggest that the stability of fasting glucose and insulin, and instability of fasting TG, LPL. Similar instability was shown in circulating levels of GPIHBP1 and HTGL at baseline.”

Revised: “These results suggested that among subjects within physiologically normal range of lipid metabolism, eating behavior affected fasting levels of TG and LPL, but not glucose and insulin. Similar effect was shown in circulating levels of GPIHBP1 and HTGL at baseline.”

  1. In Japan, KGM is commonly incorporated into various food products, however, according to our unpublished data, there are seasonal fluctuations in the intake of KGM by the Japanese. Many in winter and few in summer. This study was conducted in the summer. Therefore, it is not necessary to consider daily intake of KGM in this study. We added these sentences in Discussion (line 334-336 page 9).

“Fifth, in Japan, KGM is commonly incorporated into various food products, however, according to our unpublished data, there are seasonal fluctuations in the intake of KGM by the Japanese. Many in winter and few in summer. This study was conducted in the summer. Therefore, it is not necessary to consider daily intake of KGM in this study.”

  1. Finally, due to limited funding for this research, we could not control eating behavior. Abundant research funding is needed to control eating behavior. Please understand our funding situation.

We added “This study follows the rules of the Declaration of Helsinki.” in 2.1. Participants (line 88 page 2).

Reviewer 2 Report

  1. The work presented for review is interesting, although in my opinion the study was not properly planned. The research sample is too small (only thirteen people). It is impossible to indicate any reliable regularities on such a small research sample. We can indicate some tendency, but these will not be conclusions. The survey should be extended to include a larger group of people taking part in the research. Moreover, the supply of the gruel with KGM and, more precisely, the time when people consumed this gruel is also questionable. In my opinion, there are too many other variables resulting from the diet that can influence the results of the measured parameters. Which of course is not possible to capture in this study. Thus, the indication that the gruel was consumed on Sunday once a day for three weeks is insufficient to obtain reliable results allowing to conclude on the effectiveness of KGM. I believe that the study should be more thoughtful in the area of determining the supply of gruel with KGM and that more people should be included in the study. The diet should be the same or similar. Fewer variables, more reliable, precise and accurate results.
  2. Also, in my opinion, the submitted work fits as a short communication rather than an article.
  3. What guided the authors to adopt the following protocols? (Figure 1)
  4. Small mistake 0.8&G – should be 0.8%G – line 144 page 4.

Author Response

  1. The work presented for review is interesting, although in my opinion the study was not properly planned. The research sample is too small (only thirteen people). It is impossible to indicate any reliable regularities on such a small research sample. We can indicate some tendency, but these will not be conclusions. The survey should be extended to include a larger group of people taking part in the research. Moreover, the supply of the gruel with KGM and, more precisely, the time when people consumed this gruel is also questionable. In my opinion, there are too many other variables resulting from the diet that can influence the results of the measured parameters. Which of course is not possible to capture in this study. Thus, the indication that the gruel was consumed on Sunday once a day for three weeks is insufficient to obtain reliable results allowing to conclude on the effectiveness of KGM. I believe that the study should be more thoughtful in the area of determining the supply of gruel with KGM and that more people should be included in the study. The diet should be the same or similar. Fewer variables, more reliable, precise and accurate results.
  2. Also, in my opinion, the submitted work fits as a short communication rather than an article.
  3. What guided the authors to adopt the following protocols? (Figure 1)
  4. Small mistake 0.8&G – should be 0.8%G – line 144 page 4.

Answer: We deeply appreciate the reviewers' comments. We would like to explain our opinion. Please confirm our answers.

  • The reason of the research sample is thirteen people.

Answer: At the beginning of this study, 24 participants without diabetes mellitus were included. Therefore, the previous study was performed by using the data of 24 participants (ref. 8). We reanalyzed remaining samples from all of 24 participants. We determined the circulating levels of TG, FFA, LPL, GPIHBP1 and HTGL among 24 participants. Then, we submitted the manuscript using the data of 24 participants to Nutrients at the end of 2020(Manuscript ID: nutrients-1078748). Unfortunately, our manuscript was not accepted, however, a reviewer suggested us to reanalyze our data. The reviewer recommended us an exclusion of subjects with dyslipidemia from 24 participants. Among 24 participants without diabetes mellitus, 11 participants were excluded because of dyslipidemia. Therefore, we reanalyzed the data of 13 participants without dyslipidemia in this study. We agree with the reviewer’s opinion, therefore, we described “First, only a small number of subjects were examined.” as a limitation of this study in Discussion (line 327 page 8).

  • The supply of the gruel with KGM and, more precisely, the time when people consumed this gruel is also questionable.

Answer: All participant ate rice gruel at the same time. Because of the difficulty of dissolving KGM powder into rice gruel, the manufacturer (GREEN LEAF Co., Ltd., Showa, Gunma, Japan) developed the original method to make rice gruel containing KGM powder. Each rice gruel, 0%G, 0.4%G and 0.8%G was packed exactly 250g by the manufacturer. Therefore, each rice gruel, 0%G, 0.4%G and 0.8%G was same for all participant. To explain these facts, we added several sentences as described below.

  • We added the sentence described as follows in 3 Study design (line 108-110 page 3).

“All participants performed a rice gruel tolerance test after a 12-h overnight fast. All participants started eating rice gruel at 8:40 and ate within 10 minutes.”

  • We added the sentence described as follows in 2 Gruel test (line 99-103 page 3).

“Because of the difficulty of dissolving KGM powder into rice gruel, the manufacturer (GREEN LEAF Co., Ltd., Showa, Gunma, Japan) developed the original method to make rice gruel containing KGM powder. Each rice gruel containing 0%, 0.4% and 0.8% of KGM powder was packed exactly 250g by the manufacturer.”

  • There are too many other variables resulting from the diet that can influence the results of the measured parameters. Which of course is not possible to capture in this study. Thus, the indication that the gruel was consumed on Sunday once a day for three weeks is insufficient to obtain reliable results allowing to conclude on the effectiveness of KGM. I believe that the study should be more thoughtful in the area of determining the supply of gruel with KGM and that more people should be included in the study. The diet should be the same or similar.

Answer:

  • We performed the rice gruel tolerance test after a 12-h overnight fast. We did not control daily eating behavior. Although we did not control eating behavior, fasting levels of glucose and insulin showed stable and narrow range (our previous study, ref. 8), while fasting levels of TG, LPL, GPIHBP1 and HTGL showed wide range. Both our previous study and this study analyzed the same blood sample. These results suggested that among subjects within physiologically normal range of lipid metabolism, eating behavior affected fasting levels of TG and LPL, but not glucose and insulin. Similar effect was shown in circulating levels of GPIHBP1 and HTGL at baseline. To expand these discussion, we added the sentence described as follows in Discussion (line 287-290 page 8).

The former sentence: “These results suggest that the stability of fasting glucose and insulin, and instability of fasting TG, LPL. Similar instability was shown in circulating levels of GPIHBP1 and HTGL at baseline.”

Revised: “These results suggested that among subjects within physiologically normal range of lipid metabolism, eating behavior affected fasting levels of TG and LPL, but not glucose and insulin. Similar effect was shown in circulating levels of GPIHBP1 and HTGL at baseline.”

  • In Japan, KGM is commonly incorporated into various food products, however, according to our unpublished data, there are seasonal fluctuations in the intake of KGM by the Japanese. Many in winter and few in summer. This study was conducted in the summer. Therefore, it is not necessary to consider daily intake of KGM in this study. We added these sentences in Discussion (line 334-336 page 9).

“Fifth, in Japan, KGM is commonly incorporated into various food products, however, according to our unpublished data, there are seasonal fluctuations in the intake of KGM by the Japanese. Many in winter and few in summer. This study was conducted in the summer. Therefore, it is not necessary to consider daily intake of KGM in this study.”

  • Finally, due to limited funding for this research, we could not control eating behavior. Abundant research funding is needed to control eating behavior. Please understand our funding situation.

  1. Also, in my opinion, the submitted work fits as a short communication rather than an article.

Answer: It is impossible to reduce our data and discussion. Please understand our situation.

  1. What guided the authors to adopt the following protocols? (Figure 1)

Answer: Previous reports showed positive effects of KGM by long term habitual ingestion of KGM. We intended to investigate immediate effects of KGM on postprandial TG metabolism in this study. Therefore, we performed rice gruel tolerance test after a 12-h overnight fast. To explain the purpose of this study, we described several sentences in Introduction (line 47-51 page 2). “Previous studies have shown that KGM offers major health benefits—such as lower blood glucose, cholesterol, triglyceride (TG), and blood pressure levels as well as reduced body weight—by promoting intestinal activity and boosting immune function in humans [1-7]. In these investigations, participants habitually ingested KGM every day for 8 weeks [1-7]”

Introduction (line 57-64, page 2), “only a few studies have demonstrated its impact on immediate suppression of postprandial glucose, insulin, and lipid levels in individuals with normal or impaired glucose tolerance. Our previous study was the first to show that the intake of KGM-supplemented rice gruel could suppress postprandial increase in circulating glucose and insulin levels [8]. Expanding our previous research, we investigated immediate effects of KGM on postprandial TG, their associated enzyme called LPL, and a related glycolipid-anchored protein known as glycosylphosphatidylinositol-anchored high-density lipoprotein-binding protein 1 (GPIHBP1).”

  1. Small mistake 0.8&G – should be 0.8%G – line 144 page 4.

Answer: We appreciate the kindly comment of the reviewer. We revised “0.8&G” as ”0.8%G” line 151 page 4.

  1. We added “This study follows the rules of the Declaration of Helsinki.” in 2.1. Participants (line 88 page 2).

Round 2

Reviewer 1 Report

OK with the changes in the manuscript

Reviewer 2 Report

The paper can be accepted in the present form.